# Animal Welfare Science: Why and for Whom?

**DOI:** 10.3390/ani13111833

**Published:** 2023-06-01

**Authors:** Alessandra Akemi Hashimoto Fragoso, Karynn Capilé, Cesar Augusto Taconeli, Gabrielle Cristine de Almeida, Paula Pimpão de Freitas, Carla Forte Maiolino Molento

**Affiliations:** 1Animal Welfare Laboratory, Federal University of Paraná, Rua dos Funcionários, 1540, Curitiba 80035-050, Brazil; hashimoto@ufpr.br (A.A.H.F.); karynn.capile@gmail.com (K.C.);; 2Department of Statistics, Federal University of Paraná, Rua Cel. Francisco Heráclito dos Santos, 100, Curitiba 81531-980, Brazil; cetaconeli@gmail.com

**Keywords:** animal ethics, farm animal, motivation, sentience, welfarism

## Abstract

**Simple Summary:**

Considering that there are many ways to approach animal welfare, we aimed to study the value attributed to the animals themselves in scientific papers about farm animals published in animal welfare (AW) and animal production (AP) journals over time. All the papers were analyzed by five assessors, resulting in moderate agreement because the language used in the animal welfare scientific literature is ambiguous in relation to why and for whom it is performed. The overall mean score was 5.60, showing a low consideration of the interest of animals, close to neutrality. While AW journal scored higher and improved over the decades, the AP average score was below 5.0 and did not improve with time. The statement of the main justification for animal welfare papers, with an explicit declaration of their motivational priorities is important for the improvement of animal welfare science.

**Abstract:**

There are, in the literature, distinct ways to approach animal welfare. The objective of this work was to study the value attributed to farm animals in the scientific papers published in animal welfare and animal production journals at three different points in time, separated by a decade each. The first ten papers mentioning “animal welfare” or “animal well-being” in their objectives or hypotheses from each journal and each focus year were selected. The 180 papers were blindly scored by five assessors between 1 and 10, according to the degree of intrinsic value attributed to animals. The overall mean score and standard deviation were 5.60 ± 2.49, with 6.46 ± 2.29 and 4.74 ± 2.40 for AW and AP journals, respectively, and 5.37 ± 2.44, 5.68 ± 2.52 and 5.75 ± 2.41 for the focus years of 2000, 2010 and 2020, respectively. There was an interaction between focus year and publication area: papers from AW journals scored better over time, in contrast with papers from AP journals, for which scores remained similar over decades. The inter-assessor agreement is moderate, which may reflect the subject complexity, as the language used in the papers studied was ambiguous in relation to why and for whom it is performed. The low overall mean score evidenced that the animal welfare scientific publications are, on average, not prioritizing the interests of the animals. Thus, our results evidenced the presence of animal welfarism in animal welfare science, a problem that seems not to be intrinsic to animal welfare science itself, but rather to the way research is frequently conceived, conducted, interpreted, summarized and applied. Therefore, it seems urgent to further study the motivation for animal welfare research. The statement of the main justification for animal welfare papers, with an explicit declaration of the motivational priorities that constitute each scientific animal welfare study, may be an interesting recommendation for the improvement of animal welfare science.

## 1. Introduction

Animal welfare is a multifaceted concept that has evolved over time to address concerns about the treatment of animals, including, for example, the genuine concerns with animal suffering, registered in the book *Animal Machines: The New Factory Farming Industry* by Ruth Harrison (1964). The author reported the inhumane conditions in which animals were kept in intensive farming and highlighted the physical and emotional suffering experienced by these animals [1]. The book played a significant role in raising public awareness about animal welfare issues in the farming industry and was influential in shaping the establishment of animal welfare research. In the following decades, animal welfare emerged as a new scientific field that encompasses a wide range of disciplines, including veterinary medicine, animal behavior, psychology, and ethics [2]. Its development has been driven by a growing recognition that animals have the capacity to experience pain and emotions and that their welfare is a relevant ethical and moral concern [3].

While animal welfare science has made important advances in improving the treatment of animals, there are critiques that it does not necessarily view animals as beings with intrinsic value [4]. For instance, animal rights theorist Gary Francione criticizes animal welfare as an inadequate approach to dealing with the fundamental issues of the animal agriculture industry [5]. In fact, animal welfare has become an issue that attracts different interests, some of which may be even far from an animal protection view, e.g., productivity, added value to animal products, marketing, ESG compliance, animal farming social license and the economy, among others. Similarly, adherence to animal welfare standards may be motivated by a variety of factors. Legal requirements, for example, may compel businesses to implement animal welfare practices to avoid legal consequences [6]. Social compliance expectations of customers may drive businesses to prioritize animal welfare [7]. Environmental concerns may also motivate individuals to follow animal welfare standards, recognizing the negative impacts of animal agriculture on the environment [8]. Economic benefits, such as increased productivity and profitability, can also motivate businesses to prioritize animal welfare [9]. Finally, ethical considerations and a sense of responsibility towards animals may also play a significant role in motivating individuals to follow animal welfare standards [10,11].

Thus, the drive to conduct animal welfare research may be based on a wide range of motivations. Such motivations are necessarily linked to specific ethical views, with varying degrees of recognition of the intrinsic value of animals. In addition, it is not usual for authors in animal welfare science to offer an open statement regarding the motivation for each study, with the main motivation often remaining undeclared. On the other hand, science is not necessarily value-free, as the ideology and political beliefs of the scientists conducting the research shape the way research questions are framed, data is collected and analyzed, findings are interpreted, and main conclusions are built and disseminated [12]. Furthermore, the funding sources and priorities of research institutions and organizations are often guided by political and economic considerations, which can influence the direction of research and the interpretation of its results [13]. Finally, the extant praxis within a field also frequently determines how peers review scientific writing, which often makes it difficult and slow for new worldviews to enter the scientific literature. For animal welfare science this may be especially relevant, due to the heavy legacy of the cartesian approach to animals as non-sentient biological machines. Even though animal sentience is recognized by current scientific literature for vertebrates and increasingly so for invertebrates [14], papers prioritizing animal feelings may be received with varying levels of acceptance by different scientific journals [15].

Considering the variety of motivations and the distinct approaches to animal welfare according to the context and particular interests involved, the objective of this work was to study the value attributed to farm animals in the scientific papers published in animal welfare and animal production journals in three different points in time, separated by a decade each.

## 2. Materials and Methods

### 2.1. Scientific Paper Selection

We compared papers published in scientific journals organized in two groups, according to main scope, denominated animal welfare (AW) and animal production (AP) journals and publication in three different decades. To compare the journal groups, we selected three AW journals and three AP journals. To assess how animals were approached over time, we selected three publication years of reference: 2000, 2010 and 2020. Journal selection impact factor (IF) was also considered, and the requirement was an IF equal to or greater than 1.1 to ensure a balance between journal quality and the number of journals available per group. The selected journals were *Journal of Applied Animal Welfare Science* (IF: 1.122), *Animal Welfare* (IF: 1.550) and *Applied Animal Behaviour Science* (IF: 2.187) for the AW group; and *Livestock Science* (IF: 1.700), *Poultry Science* (IF: 2.659) and *Journal of Dairy Science* (IF: 3.333) for the AP group, according to IF information available on 14 December 2020.

### 2.2. Data Gathering

The dataset was composed of the first 10 papers mentioning “animal welfare” or “animal well-being” in their objectives or hypotheses, starting the search with the first issue published by each journal in each focus year and ending when 10 papers were selected. The journals were accessed starting with the focus year up to five years later, except for 2020. If, even so, it was not possible to select 10 papers, the year immediately preceding the focus year was accessed as well. The dataset was complete when 10 papers were selected for each of the six journals and three reference years, with a total of 180 papers. From AW journals, only the papers about farm animals were selected, to maintain consistency with the papers from AP journals in terms of the context of animal studies. All the data was organized in a worksheet, with columns for the paper title, journal title, year of publication, abstract, studied animal species, objective statement and conclusion. Some papers presented very long and non-explicit conclusions, so we had to select the essential information to facilitate the assessment. For the blind rating of each paper, a new coded worksheet was created, containing the paper title, abstract, objective and conclusion, and omitting the journal title. The papers, which were initially entered following a chronological publication order by journal, were randomly re-ordered. The publication year or other date markers that eventually appeared on abstracts, conclusions and objectives were replaced by four asterisks (****). This coded worksheet was then used independently by five assessors to rate each paper.

### 2.3. Pilot Phasefor the Definition of Criteria and Assessment Method

Before assessing the study dataset, a short list of criteria was defined and three pilot assessments were run to refine it. The first pilot worksheet included ten papers and the following two included six papers each; none of such papers were in the study’s main database, as they were selected from different sources. For the pilot assessment results, to keep track of the coherence across scores given by the five assessors evaluating independently the same set of papers, Cohen’s kappa coefficient [16] was used and remained in the moderate range after the three rounds of paper assessment. Meetings for the discussion and improvement of criteria were required for the three pilot assessments, after which the five assessors reached a consensus for the list of criteria to be included and their respective weighted scores. It was decided that each paper was to be scored between 1 and 10, according to the established criteria (Figure 1). Each paper was scored considering all criteria, and scores were added for intrinsic value criteria and subtracted for the opposite criteria in a cumulative manner. A score of 10 was given to papers assigning intrinsic value to animals; in other words, those in which the focus was animal welfare considering the perspective of the animals. The score 1 was given to papers on exactly the opposite side of the spectrum, focusing on issues not related to the interests of the animals, such as increases in productivity, a better quality of animal products, and better reproductive success, among others, and the perspective and interests of the animals themselves were only tangentially addressed, if addressed at all. The assessment starting score for each paper was 5, the intermediate point in the scale, and then score points were added or subtracted, depending on the identification of intrinsic, neutral or extrinsic values to animals (Figure 1). 

### 2.4. Statistical Analysis

Lin’s concordance correlation coefficient [17] was used to measure the inter-rater reliability regarding the 180 papers assessment. In this case, values next to +1 (−1) represent near-perfect concordance (discordance), whereas zero denotes the complete inter-rater reliability absence.

To evaluate the effects of the journal group and decade of publication on the score of the papers regarding their approach to animal welfare, we proceeded with regression analysis. A linear mixed regression model [18] was specified with the following components: the fixed effects of journal group, decade of publication and animal species; and the random effects of assessor and scientific journal. We assumed independent normal random effects and a normally distributed random error. In addition to considering the main effects of journal group and decade of publication, we have also investigated the corresponding interaction effect to assess possible differences in the assigned scores between journal groups in each decade, and between decades into the journal groups. The results provided by the fitted model are summarized through estimated marginal means and 95% confidence intervals. Residual analysis was performed to evaluate the goodness of the fitted model. Significance tests had their *p*-values adjusted for multiple comparisons, to ensure a global significance level of 5%. 

All analyses were conducted in the R environment for statistical computation, version 4.0.2 [19]. The R library lme4 [20] was used to fit the linear mixed model.

In addition to the planned comparative statistical analysis, to measure how prevalent farm animal welfare papers were in each journal over time, we registered the required publication period and the number of papers checked until it was possible to reach the goal of 10 papers according to the selection criteria, for each focus year and journal.

## 3. Results

### 3.1. Data Gathering

The number of consulted papers and publication period required to gather the sample of 180 papers are shown in Table 1. Our initial objective was to evaluate the focus years of 2000, 2010 and 2020 to understand the time effect in a comparative way across three decades. However, it was necessary to enlarge the focus years’ representative time period to gather the expected sample of 10 papers per focus year per journal. Even though AP journals published more papers per year, a longer timeframe was necessary to gather the papers according to the selection criteria. This was expected as the AP journals are not specific to animal welfare research. On the other hand, eventually, more than a single year was required for AW journals because we were only interested in farm animal studies, and many papers in this journal group approached the welfare of animals in other contexts. An overall trend for publishing more papers addressing farm animal welfare per year was observed, as the number of years required to reach ten papers decreased consistently across decades in both journal groups, as well as the number of required individual papers.

### 3.2. Data Assessment

The statistical analysis of the main dataset showed an interaction effect between a journal’s decade of publication and publication area (*p* = 0.03) and species (*p* < 0.001) on the scores attributed. The fitted mean score for each publication area and the decade of publication were summarized in Table 2. We can observe that the scores attributed to AW journals papers were statistically higher compared with AP journals papers scores.

The interaction effect between the journal group and the decade of publication is detailed in Table 3.

Papers included in this study were those reporting research with farm animals, which resulted in a dataset with a variety of animal species, which in turn were classified into five main categories. The first category is poultry, including laying hens and broiler chickens, with 49 papers; the second one is cattle, with most papers approaching dairy cattle, with 53 papers; the third one is pigs, grouping papers about sows and piglets, including 28 papers; the fourth category groups papers reporting farm animals in general, as some papers did not approach a single species—for example, we analyzed a paper about poultry and pigs, with 28 papers; and the last category is “others”, grouping papers about less prevalent species in our sample, such as rabbits, fishes, visions and small ruminants, including 22 papers. 

In Table 4 we summarize the estimated marginal means for the main effect of animal species, aiming to understand if the animal species approach affects the consideration of animals for their intrinsic value. Table 5 presents the estimated contrast between animal species and their corresponding *p*-values.

### 3.3. Inter-Assessor Reliability

Lin’s concordance correlation coefficient was used to compare the reliability between each pair of assessors. The greater the value, the greater the reliability. In this case, zero indicates random assessments, while 1.00 represents the ideal reliability. All the values observed are in Table 6. The highest value found was 0.51, which represents intermediate reliability; three values were lower than 0.40, characterizing low reliability; and no very low-reliability value was observed.

After all the papers were assessed, the coded worksheet was decoded and the average scores were calculated. The overall mean scores and standard deviations were 5.60 ± 2.49, with 6.46 ± 2.29 and 4.74 ± 2.40 for AW and AP journals, respectively, and 5.37 ± 2.44, 5.68 ± 2.52 and 5.75 ± 2.41 for the focus years of 2000, 2010 and 2020, respectively.

## 4. Discussion

Our study aimed to investigate the value attributed to farm animals in scientific articles published in animal welfare and animal production journals at three distinct time points, each separated by a decade. Considering that our assessment measured in scores the level of consideration of the intrinsic value of animals in each paper, i.e., how much the quality of life as experienced by the animals was central to each paper, the overall average of 5.60 ± 2.49, which is close to our neutral value, is lower than expected. As the main criterium for paper inclusion in the dataset was the term “animal welfare” in either its title, objective or hypotheses, the low consideration of the intrinsic value of animals seems to contradict the history of animal welfare science as well as most accepted scientific concepts of animal welfare. Animals were the center of Harrison’s seminal book *Animal Machines* [1]. Animals are at the center of the animal welfare concept by Broom (1986), in which animal welfare refers to the state of animals as regards their attempts to cope with their environment [21]. Similarly, Mellor (2016) defines animal welfare as the quality of life that an animal experiences, encompassing its physical, emotional and psychological well-being [22]. Webster (2006) provides a simple definition of well-being as “fit and happy” or “fit and feeling good”, which explicitly refers to both the body and the mind in a state of sustained health and an absence of suffering. Feeling good should also include comfort, companionship and security [23]. Dawkins (2008) describes animal welfare as the overall state of an animal’s health, happiness and well-being, also considering that it is influenced by its environment, nutrition and social interactions [24]. Finally, Fraser et al. (2008) explain that animal welfare is a complex concept that refers to the physical, physiological and behavioral health of animals, as well as the satisfaction of their natural needs and the avoidance of suffering [25]. Thus, it seems unquestionable in the literature, considering major scientists in the history of animal welfare science, that animal welfare is centered on animals by the very definition of the term.

The proportion of scientific publications which presented a neutral or even absent focus on the animals themselves raises an important and urgent concern, as it suggests a distortion of the animal welfare concept with major implications. This concern has been reported in less scientific scenarios. For instance, marketing and agri-business discourses tend to appeal to animal welfare in their promotional strategies to improve public perception or attenuate criticism towards animal-based products [26,27]. The animal industry rhetoric of denial of animal suffering has been described in detail [28]. This can be done by strategies that aim to hide the connection between the animal product and the sentient animal from which it originates, a phenomenon known as the absent referent [29]. It can also be achieved through various forms of denial that have been described as communication strategies to promote a product or service as being environmentally friendly, humane or welfare conscious, even when this is not necessarily the case [7,30]. Examples of these strategies are narratives such as meat washing and humane washing, the latter referring to misleading statements either by describing higher animal welfare states than the reality or by installing practices and codes that lead to such misperceptions. Humane washing has been described as a type of whitewashing, which is a metaphor for communications that gloss over or obscure unpleasant, negatively connoted facts [31]. For instance, the comparison of organic certified and non-certified broiler chicken farms revealed no differences when the welfare of the animals was measured through a complete assessment, due to the low welfare standards required for certification [32]. These are examples of concerns related to the distortion of the animal welfare concept in philosophy, marketing and certification. Our results show for the first time that a similar concern applies to the scientific practices in the field of animal welfare.

The results suggest the relevance of a deeper discussion of animal welfare science within the domain of epistemology. The consideration of the overall average score reveals an uncertainty on what sort of linkage there is between ethical demands and the objectives of animal welfare studies in peer-reviewed scientific publications, with the complicating aspects of undeclared driving motivations and vague use of the term animal welfare. This truncated linkage may reflect a broader ongoing debate on the relationship between science and moral philosophy. Important authors have highlighted the imbrication between science and values, stressing that bridging these fields is epistemologically constructive for society [33,34]. It has been 20 years now since the animal welfare scholar David Fraser stated that, since the 1970s, scientists and philosophers have sought to understand our relationship with animals, but their differing concepts and vocabulary have created a divide [35]. On the other hand, it is recognized that animal welfare studies must bridge science and ethics for the best development of both [36,37,38].

The different scores between AP and AW journal groups likely relate to the different main scopes of each journal group, which may function as a stronger underlying driving principle than the inclusion of animal welfare in the objective of each paper. For instance, the scope of the AP journal *Livestock Science* is to promote “the sound development of the livestock sector by publishing original, peer-reviewed research and review articles covering all aspects of the broad field of animal production and animal science. The journal welcomes submissions on the avant-garde areas of animal genetics, breeding, growth, reproduction, nutrition, physiology, and behavior in addition to genetic resources, welfare, ethics, health, management and production systems” [39]. On the other hand, the AW journal *Animal Welfare* “publishes the results of peer-reviewed scientific research, technical studies, surveys and reviews relating to the welfare of kept animals (e.g., on farms, in laboratories, zoos and as companions) and of those in the wild whose welfare is compromised by human activities. Papers on related ethical and legal issues are also considered for publication” [40]. Thus, a statistical difference in scores between AP and AW journal groups was expected by the methodological design and confirmed by the results.

An increasing score average over the decades was expected in the face of the notable increase in academic production related to animal welfare science in the past decades, including a growing awareness of animal welfare issues and interest in understanding the science behind the welfare of animals in agriculture settings [41]. Such interest in the understanding of animal welfare has led to an influx of research and academic publications aiming to understand animal behavior, cognition and emotions, as well as to develop better methods of animal welfare assessment. However, the mean score increase over time seems modest in light of evident promotion in terms of scientific works and the social and ethical relevance of the subject [42]. The fact that for the AP journal group, the average scores did not increase over time, suggests the existence of specific factors or biases in this journal group [43,44,45].

The difficulty in assessing the motivations for each scientific study became evident as successive meetings among assessors with the goal of refining criteria for scoring the papers in terms of the level of consideration of the intrinsic value of the studied animals were needed. Thus, the reason for studying animal welfare was not evident in most papers studied. In addition, after three pilot analyses, the level of agreement among assessors remained only moderate, according to the kappa coefficient. The concordance among assessors remained only moderate for the assessment of the papers in our main dataset as well; however, even under such score variance, significant differences related to our research questions were observed. The lack of clear statements regarding the motivation for animal welfare studies or of an easily understandable motivational context suggests a need for improvement in scientific writing in the animal welfare field, as more clarity is preferable [46]. The proportion of authors who are completely self-aware of their own motivations, as well as the eventual level of such awareness, are interesting research questions that warrant further studies. The uncertainty in terms of self-awareness also indicates that additional thought and a more declarative writing of the reasons driving each animal welfare paper may be beneficial to the development of this scientific area and its potential contribution to society.

The number of years and of papers reviewed before reaching the ten required papers per journal in each decade showed some interesting patterns. First, for most journals except *Animal Welfare*, the number of years required decreased with the decade; this suggests that papers approaching animal welfare became more prevalent in AP journals and that papers approaching farm animals became more prevalent in AW journals. Such results seem to show a similar trend of the increasing value for farm animal welfare issues given by society in general, which agrees with the view that animal welfare is a mandated science driven by societal demands. The term mandated science describes research commissioned to guide actions, decisions and policies related to concepts such as welfare, which consist of both scientific and value-based considerations [24]. Such a parallel increase in attention to farm animal welfare in both societal demands and in scientific publications suggests the relevance of a similar coherence regarding motivation and impact of the knowledge thus generated. A discrepant, ambiguous or non-declarative approach within animal welfare science may risk an alignment or the fueling of practices discussed in terms of animal product marketing, including, for example, humane washing and meat washing. Meat washing refers to rhetorical discourses which obscure animal feelings, especially animal suffering, within the context of propaganda for food of animal origin [47].

In the field of human-animal studies, speciesism is a growing area of study, and language is recognized as an important form of its manifestation [48,49,50]. The lack of significance for the comparison of scores across species in our results is likely a consequence of the scope adopted for sampling, where only animal species used for farming purposes were considered. As the moral status of animals socially labeled as farm animals or seen as food is similar, namely, commodities [44,51], no difference in scores among the species studied was expected, as they fall within the context of animals farmed for food production. In this sense, further research on the motivations for animal welfare scientific studies involving an array of animal use contexts seems warranted, as they may show how intense the phenomenon here described is in different contexts of animal use and, thus, contribute to the understanding of underlying reasons for the weak consideration of the intrinsic value of animals in farm animal welfare research. 

The expanding investigation area on the psychology of interspecies relationships gives a conceptual theoretical framework capable to describe and explain many tendencies and biases in relation to the form that human groups with different interests understand and approach animals [43]. This area of study describes certain phenomena involving cognitive dissonance, defense mechanisms and moral justification that have been extensively studied, such as mind denial [52], the meat paradox [53], the 4Ns [54] and moral disengagement [55], which may contribute to further the understanding and the relevance of our results. In this line, research has shown that people’s attitudes toward animals are shaped by a complex interplay of factors, including cultural and social influences, personal experiences and cognitive biases. For example, studies have observed that people who consume meat are more likely to exhibit attitudes that justify animal use, such as beliefs that animals do not have feelings or that their suffering is necessary for human survival [52,56]. On the other hand, exposure to animals and education about their cognitive and emotional abilities has been shown to increase empathy and concern for animals [57]. How such underlying phenomena shape animal welfare research questions and the language choice in scientific writing, as well as the weight allowed for the consideration of the interests of the animals, is an underexplored area of research. An additional issue that seems to warrant further studies is the effect of the geographical origin of the papers, as regional cultural factors may differently impact the motivations for research in the field of animal welfare.

The difficulties in developing criteria for paper assessment, as well as the moderate concordance among assessors and the relatively high standard deviation for the overall mean score, seem to originate from the fact that the moral consideration of animals by the authors was frequently uncertain, ambiguous, or undeclared, which suggests that animal welfare science is not necessarily focused on the interest of animals, as the concept intuitively imply. This finding is reinforced by the high frequency with which motivations other than the intrinsic value of animals were explicitly stated. Animal welfare science may lose public support and, consequently, the very reason for its existence if it is not centered on animals. In fact, the validity of animal welfare science has been questioned in philosophy, through the concept of animal welfarism [58]. This may lead to the endorsement of the perception that animal welfare cannot contribute to improving the possibilities for better lives for animals, hindering positive changes for animals that are dependent on knowledge produced by animal welfare science [15]. In this sense, we stress the importance of clarifying the ethical framework for animal welfare in scientific writing, to ensure transparency and credibility towards society. That way, the bases of moral consideration of animals in the field of animal welfare science become explicit, dissipating the difficulties for their identification which are prevalent in farm animal welfare papers, as demonstrated by our work.

## 5. Conclusions

Our methodological challenges and results warn about the complexity of the subject addressed in this study. The language used in the animal welfare scientific literature is ambiguous in relation to why and for whom it is performed. Overall, the absence of consideration of the interests of animals is surprisingly frequent for animal welfare papers, in disagreement with the scientific definition of the term animal welfare. The papers from AW journals tended to show a more animal-centered position which improves over decades; on the other hand, the animal welfare literature from AP journals approached animals in a more tangential way, often prioritizing other reasons for improving animal welfare than the animals themselves, with no improvement in the last decades. Thus, our results evidenced the presence of animal welfarism in animal welfare science, a problem that seems not to be intrinsic to animal welfare science itself, but rather to the way research is frequently conceived, conducted, interpreted, summarized and applied. Therefore, it seems urgent to further study the motivation for animal welfare research. The statement of the main justification for animal welfare papers, with an explicit declaration of the motivational priorities that constitute each scientific animal welfare study, may be an interesting recommendation for the improvement of animal welfare science. 

## Figures and Tables

**Figure 1 animals-13-01833-f001:**
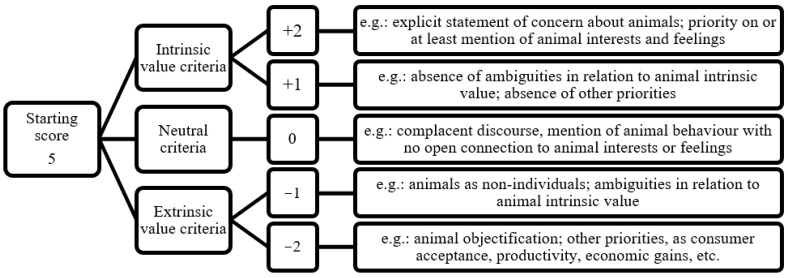
Criteria for the analysis of animal centrality on 180 scientific papers, with cumulative addition and subtraction of scores, performed from September to December 2020.

**Table 1 animals-13-01833-t001:** Number of consulted papers and time frames required to gather the 180 papers database for the analysis of animal welfare approach, considering the established criteria of keywords (animal welfare, animal well-being) and only farm animal studies.

*Journal Title*, Journal Group	Timeframe for Each Focus Year per Journal	Number of Years Required	Number of Checked Papers
*Animal Welfare*, Animal Welfare	2000 to 2002	3	160
	2010	1	79
	2019 to 2020	2	101
*Journal of Applied Animal Welfare Science*, Animal Welfare	1999 to 2005	7	242
	2010 to 2013	4	144
	2019 to 2020	2	77
*Applied Animal Behaviour Science*, Animal Welfare	2000 to 2003	4	571
	2010 to 2011	2	359
	2020	1	170
*Livestock Science*, Animal Production	2000 to 2003	4	635
	2010 to 2012	3	988
	2019 to 2020	2	704
*Journal of Dairy Science*, Animal Production	1999 to 2005	7	2956
	2010 to 2011	2	1358
	2020	1	1100
*Poultry Science*, Animal Production	2000 to 2004	5	1435
	2010	1	352
	2020	1	878

**Table 2 animals-13-01833-t002:** Fitted estimated means for scores regarding animals as primary motivation for each journal group and decade, as per the assessment of 180 scientific papers from animal welfare (AW) and animal production (AP) journals.

Decade	Journal Group, Number of Papers	Estimated Means (Standard Error)	95% Confidence Interval
2000	AP, 30	4.59 (0.44)	(3.52; 5.65)
	AW, 30	6.09 (0.44)	(5.03; 7.15)
2010	AP, 30	4.90 (0.44)	(3.83; 5.96)
	AW, 30	6.27 (0.44)	(5.21; 7.34)
2020	AP, 30	4.56 (0.44)	(3.50; 5.63)
	AW, 30	6.84 (0.44)	(5.77; 7.90)

**Table 3 animals-13-01833-t003:** Estimated mean contrasts for scores regarding animals as primary motivation by journal group and decade, as per the assessment of 180 scientific papers from animal welfare (AW) and animal production (AP) journals, based on the interaction effect between journal group and decade of publication.

Fixed	Contrasts	Difference (Standard Error)	95% Confidence Interval	*p*-Value
2000	AW–AP	1.50 (0.32)	(0.80; 2.21)	<0.001
2010	AW–AP	1.38 (0.33)	(0.66; 2.10)	0.001
2020	AW–AP	2.27 (0.34)	(1.54; 3.01)	<0.001
AW	2010–2000	0.18 (0.25)	(−0.41; 0.78)	0.748
	2020–2000	0.75 (0.26)	(0.13; 1.36)	0.011
	2020–2010	0.56 (0.26)	(−0.04; 1.17)	0.074
AP	2010–2000	0.31 (0.28)	(−0.34; 0.96)	0.500
	2020–2000	−0.02 (0.28)	(−0.68; 0.64)	0.997
	2020–2010	−0.33 (0.25)	(−0.93; 0.27)	0.391

**Table 4 animals-13-01833-t004:** Estimated marginal score means regarding animals as primary motivation, as per the assessment of 180 scientific papers from animal welfare (AW) and animal production (AP) journals.

Species	Estimated Mean Score (Standard Error)	95% CI
Poultry	5.74 (0.43)	(4.67; 6.81)
Cattle	5.75 (0.43)	(4.69; 6.81)
Pigs	6.06 (0.44)	(5.01; 7.12)
Farm animals	4.44 (0.45)	(3.39; 5.50)
Other species	5.71 (0.44)	(4.65; 6.76)

**Table 5 animals-13-01833-t005:** Estimated mean contrasts for scores regarding animals as primary motivation, as per the assessment of 180 scientific papers from animal welfare (AW) and animal production (AP) journals, based on the main effect of animal species studied.

Contrasts, Number of Papers	Difference (SE)	95% CI	*p*-Value
Farm animals, 28, poultry, 49	−1.30 (0.29)	(−2.10; −0.49)	<0.001
Farm animals, cattle, 53	−1.30 (0.28)	(−2.07; −0.53)	<0.001
Farm animals, pigs, 28	−1.62 (0.30)	(−2.43; −0.80)	<0.001
Farm animals, others, 22	−1.26 (0.28)	(−2.04; −0.49)	<0.001
Poultry, cattle	−0.01 (0.27)	(−0.77; 0.76)	0.999
Poultry, pigs	−0.32 (0.27)	(−1.07; 0.44)	0.770
Poultry, others	−0.04 (0.28)	(−0.73; 0.80)	0.999
Cattle, pigs	−0.31 (0.27)	(−1.04; 0.41)	0.759
Cattle, others	0.04 (0.27)	(−0.69; 0.77)	0.999
Pigs, others	0.35 (0.28)	(−0.40; 1.11)	0.703

**Table 6 animals-13-01833-t006:** Inter-assessor reliability Lin’s coefficient values for scores regarding animals as primary motivation, as per the assessment of 180 scientific papers from animal welfare (AW) and animal production (AP) journals.

	Assessor 2	Assessor 3	Assessor 4	Assessor 5
Assessor 1	0.44 (0.33; 0.55)	0.51 (0.40; 0.61)	0.48 (0.37; 0.58)	0.43 (0.31; 0.53)
Assessor 2	-	0.48 (0.36; 0.59)	0.39 (0.29; 0.48)	0.40 (0.31; 0.49)
Assessor 3	-	-	0.34 (0.22; 0.44)	0.35 (0.24; 0.44)
Assessor 4	-	-	-	0.43 (0.30; 0.54)

## Data Availability

The data presented in this study are available online at https://drive.google.com/drive/folders/12mcGUxFCOiBZfEHwsK80Ub6U7mQpI8hD?usp=sharing (accessed on 29 May 2023).

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
