# Peer review of "Animal Welfare Science: Why and for Whom?"

_animals, 2023, doi:10.3390/ani13111833_

Round 1
Reviewer 1 Report
I read your paper carefully. I think you need to revise some parts of it and please clarify what you can suggest directly on the basis of analyzing the score.
The objective indicated in the last part of Introduction is not well stated in Discussion (especially, in the first paragraph). I think discussing the objective should be on the basis of the results.
There are some mistakes on headings. 2.3 section (Pilot assessment phase) may not include only pilot assessment phase and may include method of your main research. And I can not understand how to score from the descriptions in section 2.3 and Figure 1. From the explanation of Figure1, I think scores will be ranging from 3 to 7 and for example, please explain how to score 10 points on assessed papers.
The heading of section 3.2 (Statistical analysis) is incomprehensive. Section 3.3 is absent.
>Discussion
You stated that the reason for studying animal welfare was not evident for most papers (L247-248) and the moral consideration of the animals was frequently uncertain or absent (L392-393), but I can not understand why you suggested or concluded, and which results you did on the basis of. Are the suggestion and conclusion on the basis of scoring ? ( I don't think score less than 5 can support them). Please clarify the sentences in Discussion.
Author Response
Dear Reviewer,
Please find our answers in the attached PDF. For greater clarity, the excerpts from the reviewer’s message are copied in italic.

Reviewer 2 Report
The paper is well written, and it provides interesting elements for discussion around a relevant issue, somehow also related to the misleading use of buzzwords (I.e., COVID) to attract the interest of the readers.
I have basically one comment and a suggestion:
-
A) The authors checked an outstanding number of 12309 papers to gather the sample of 180 publications. According to their approach, only the first 10 papers mentioning “animal welfare” or “animal well-being" in their objectives or hypothèses, were selected, starting with the focus-year (2000, 2010 and 2020) for up to five years later, except for 2020. As a result, the papers published from 2006 to 2009 and from 2014 to 2018 were not included. Therefore, it might be argued that the attributed scores could be not representative of the papers published on the focus-years (2000, 2010 and 2020) nor of the ones published over the time, from 2000 to 2020. The alternative approach could have been to select the first mentioning “animal welfare” or “animal well-being" published each year, from 2000 to 2020. Could the authors clarify the reason for this choice?
-
B) Whenever feasible, it would be interesting to include and discuss the interaction effect between journals decade of publication and the geographical distribution of the papers (I.e., Europe, Americas, Africa, Asia...).
Author Response
Dear Reviewer,
Many thanks for your contribution. Please find our answers in the attached PDF.

Reviewer 3 Report
This paper makes a valuable contribution by demonstrating that the motivational context of animal welfare research is often obscure and, moreover, often not actually focused on the welfare of animals. Increased clarity and transparency in animal welfare research would be of great benefit, both scientifically and ethically. The finding that most articles scored relatively low on level of consideration of animals' intrinsic value is both fascinating and disturbing.
I hope the authors continue to pursue this line of research. It would be very interesting to conduct similar analyses of different venues of animal use (animals in zoos; animals in research settings; animals kept as pets), and it would also be interesting to compare the animal welfare literature with the animal behavior and animal cognition literature.
I am not a quantitative scientist and don't feel that I am qualified to assess the research methodology and statistical analysis used in this study. I based my review on what I consider to be the intellectual merits of the inquiry and the discussion, and the value I see in broadening the scope of discussion in animal welfare science. The authors are addressing a critical perspectival blindspot in the field/ The topic is highly relevant. Yes, the references seem appropriate and well placed.
One small correction: Line 46. Animal Machines--capitalize and italicize.
Author Response
Dear Reviewer,
Many thanks for your contribution. Please find our answers in the attached PDF

Round 2
Reviewer 2 Report
Thanks for having taken on board my comments and suggestions, I hope you will be able to disseminate this paper to the benefit of animals and the quality of animal welfare research.